# Thermodynamic and structural anomalies of water nanodroplets

Shahrazad M. A. Malek [1], Peter H. Poole [2] & Ivan Saika-Voivod [1]

Liquid water nanodroplets are important in earth's climate, and are valuable for studying supercooled water because they resist crystallisation well below the bulk freezing temperature. Bulk liquid water has well-known thermodynamic anomalies, such as a density maximum, and when supercooled is hypothesised to exhibit a liquid–liquid phase transition (LLPT) at elevated pressure. However, it is not known how these bulk anomalies might manifest themselves in nanodroplets. Here we show, using simulations of the TIP4P/2005 water model, that bulk anomalies occur in nanodroplets as small as 360 molecules. We also show that the Laplace pressure inside small droplets reaches 220 MPa at 180 K, conditions close to the LLPT of TIP4P/2005. While the density and pressure inside nanodroplets coincide with bulk values at moderate supercooling, we show that deviations emerge at lower temperature, as well as significant radial density gradients, which arise from and signal the approach to the LLPT.

[1] Department of Physics and Physical Oceanography, Memorial University of Newfoundland, St. John's, NL A1B 3X7, Canada. [2] Department of Physics, St. Francis Xavier University, Antigonish, NS B2G 2W5, Canada. Correspondence and requests for materials should be addressed to I.S.-V. (email: saika@mun.ca)

Nanoscale particles of water are a key component of important processes in the earth's atmosphere, planetary and interstellar space, and numerous technology applications[1–5]. For example, nanometre-sized aqueous aerosol droplets are common in earth's lower atmosphere, and understanding their role in cloud formation is critical for climate prediction[6]. The crystallisation of pure water nanodroplets has attracted particular interest because the temperature at which freezing is observed, relative to bulk water, decreases dramatically with size, reaching 202 K for nanodroplets of radius 3.2 nm[7]. This effect arises from a combination of influences: surface effects normally lower the melting temperature of a small system relative to the bulk[8]; a smaller system volume yields fewer nucleation events[9]; and, importantly for experiments, the large surface-to-volume ratio of a small droplet makes rapid cooling rates possible, allowing the establishment of low-temperature conditions on a time scale shorter than the nucleation time[10, 11].

On cooling, bulk liquid water exhibits well-known thermodynamic anomalies, such as the density maximum at 277 K[12]. As the temperature $T$ decreases into the supercooled regime, these anomalies become progressively more dramatic. For example, both the specific heat and the isothermal compressibility of the liquid increase strongly as $T$ decreases. To account for these anomalies, several thermodynamic scenarios have been proposed, including the hypothesis that a liquid-liquid phase transition (LLPT) occurs in deeply supercooled water[13, 14]. However, bulk samples of liquid water crystallise at a homogeneous nucleation temperature $T_H$ (encountered at ambient pressure in the range 227–232 K, where the precise value depends on the experimental protocol[10, 15–18]), which to date has prevented the direct observation of the LLPT predicted to occur at lower $T$. The ability of water nanodroplets to remain liquid below $T_H$ presents a promising opportunity to clarify the properties of deeply supercooled water, provided that the bulk anomalies are not suppressed as the number of molecules $N$ in a nanodroplet decreases[7, 11, 16, 19].

In addition, as the size of water nanodroplets decreases, they access a range of pressure $P$ above ambient, due to the Laplace pressure $P_L$ that arises inside a liquid droplet. As pointed out in ref.[19], the increase of $P_L$ in small water nanodroplets also contributes significantly to the decrease of their freezing temperature. From the Young–Laplace equation $P_L = 2\gamma/R$, where $R$ is the droplet radius and $\gamma$ is the surface tension, $P_L$ inside a 1 nm droplet should exceed $10^2$ MPa[19, 20]. This is high enough to approach the range of $P$ in which the critical point of the proposed LLPT is estimated to occur in bulk water[14, 21]. Water nanodroplets thus permit exploration of a significant range of both $T$ and $P$ relevant to understanding deeply supercooled water.

Despite the importance of liquid water nanodroplets, and their potential to help clarify the behaviour of bulk water, relatively little is known of their fundamental thermophysical properties. This is due to the significant experimental challenges associated with studying liquid nanodroplets that are not in contact with a supporting or confining surface. To date, experimental and simulation studies of pure liquid water nanodroplets have focussed largely on freezing and melting behaviour[7, 8, 19, 22–28], as well as the formation of amorphous solid nanoparticles[29]. However, a systematic description is lacking for how basic nanodroplet properties, such as $R$, $P_L$, or the droplet density profile, vary with both $N$ and $T$. Knowledge of this variation is necessary to determine the regime in which bulk liquid properties, including the anomalies of bulk water, first emerge as nanodroplets grow in size. Also lacking is an understanding of how a liquid nanodroplet will behave under $T$–$P$ conditions at which the corresponding bulk liquid exhibits a LLPT.

Here we seek to address these knowledge gaps through computer simulations of water nanodroplets, modelled using the

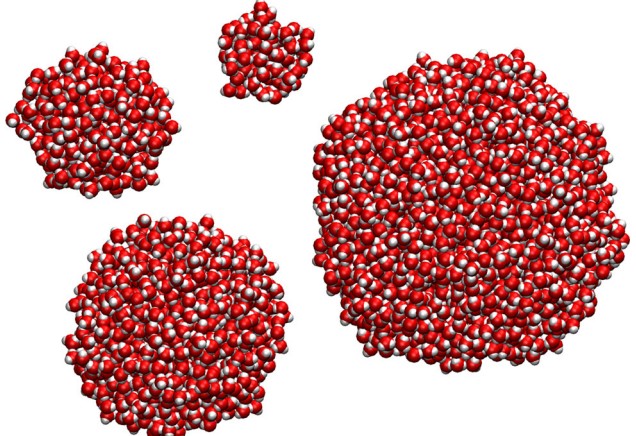

**Fig. 1** Snapshots of simulated liquid water nanodroplets. Equilibrium nanodroplets at $T = 200$ K for various sizes $N = 100$, 360, 1100, and 2880

TIP4P/2005 interaction potential[30]. The TIP4P/2005 model is known to reproduce the phase behaviour and thermodynamic anomalies of bulk water over a wide range of $T$ and $P$, and also predicts the occurrence of a LLPT with a critical point located at $T_c = 182$ K and $P_c = 170$ MPa[21]. As we will show, by comparing nanodroplet and bulk behaviour for the same water model, we self-consistently estimate the range of $N$ for which bulk properties emerge, and also identify novel nanodroplet behaviour that occurs when approaching the conditions of the bulk LLPT observed in the model.

## Results

**Anomalous variation of the nanodroplet radius.** We study isolated equilibrium nanodroplets consisting of $N$ molecules, where $N$ ranges from 100 to 2880, for $T$ from 180 to 300 K; see Methods for details of our simulations. Example nanodroplets from our simulations are shown in Fig. 1.

We characterise the nanodroplet size as a function of $N$ and $T$ by evaluating the average radius $R$, as described in Methods. If the density of droplets is constant, then $R^3$ will be proportional to $N$. In order to reveal more subtle variations in $R(N, T)$, we first define an effective droplet density as determined by $R$ as $\rho_R = 3mN/4\pi R^3$, where $m$ is the mass of a water molecule, in order to scale out the approximate proportionality of $R^3$ and $N$. Next, we note from the Young–Laplace equation that $\gamma/R$ should be proportional to $P_L$. As we will see below, we find that $\gamma$ is approximately constant at fixed $T$ over the range of $R$ studied here. Hence $R^{-1}$ should be proportional to $P_L$ along isotherms, and so $R^{-1}$ can serve as a proxy for the pressure inside a nanodroplet. We therefore present in Fig. 2 our data for $R(N, T)$ plotted as isotherms of $R^{-1}$ versus $\rho_R$, a form analogous to the equation of state (EOS) of a bulk liquid when plotted as isotherms of $P$ versus the bulk liquid density $\rho$.

The EOS of the TIP4P/2005 bulk liquid is shown in Fig. 3a, and displays several important anomalies of water[12, 21]. When an EOS is presented as isotherms of $P$ versus $\rho$, as in Fig. 3a, the occurrence of a density maximum along isobars is indicated by the crossing of isotherms. That is, if two isotherms intersect in the $\rho - P$ plane, then the density is equal at two different $T$ at the same $P$, a condition that occurs on either side of a density maximum. A maximum in the isothermal compressibility $K_T = \rho^{-1}(\partial\rho/\partial P)_T$ as a function of $P$ at fixed $T$ corresponds to the emergence of an inflection in the isotherms at the lowest $T$. Increasing $K_T$ on cooling is reflected in the decreasing slope of the isotherms as a function of $T$ at fixed $P$, and is a precursor of the divergence of $K_T$ at the critical point of the proposed LLPT.

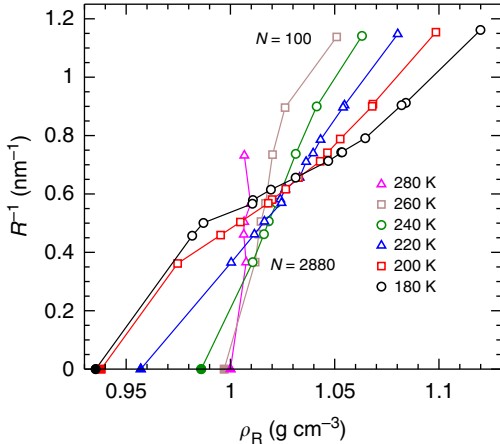

**Fig. 2** Variation of nanodroplet radius $R$ with temperature $T$ and number of molecules $N$. Isotherms of $R^{-1}$ versus the effective droplet density $\rho_R = 3mN/4\pi R^3$. The statistical error for both $R^{-1}$ and $\rho_R$ is smaller than the symbol size. $N$ decreases with increasing $R^{-1}$ along each isotherm. The filled symbols locate the bulk behaviour expected for droplets as $R \to \infty$ and $N \to \infty$.

Each of the anomalous features enumerated above for the bulk EOS is also observed in the nanodroplet isotherms derived from $R(N, T)$ and plotted in Fig. 2. That is, the nanodroplet isotherms for $R^{-1}$ versus $\rho_R$ also cross; inflect at low $T$; and exhibit a range of $R^{-1}$ in which the slope decreases as $T$ decreases. We thus find that the variation of $R$ with $N$ and $T$ exhibits the signatures of water's bulk anomalies as observed over a wide range of $\rho$ and $P$. The occurrence of this qualitative correspondence is remarkable, given that these nanodroplets are extremely small relative to a bulk system, and have no external pressure applied to them.

**Density profile of nanodroplets**. To quantify the internal structure of our nanodroplets, we study the density as a function of the distance $r$ from the droplet centre of mass. We first compute $\rho_o(r)$, the density of molecules that have their centres of mass in a shell of radius $r$, shown in Fig. 4a. As noted in previous simulations of water nanodroplets[19, 27, 29], we observe oscillations in $\rho_o(r)$ that are especially prominent near the surface, indicating that the interface with the vacuum is a well-defined molecular layer, the influence of which propagates inward as a succession of concentric shells. The amplitude of these oscillations is larger at lower $T$ and for smaller $N$.

Although the oscillations of $\rho_o(r)$ reveal the shell-like structure of nanodroplets, their large amplitude makes it difficult to define an average density for the droplet interior. As an alternative measure of the density profile, we compute the Voronoi cells for all O atoms, ignoring the H atoms. Within each shell of radius $r$, we compute the total volume $\mathcal{V}(r)$ of the Voronoi cells for O atoms, as well as $\mathcal{N}(r)$, the number of O atoms. We define the average density as determined by the Voronoi cell volumes as $\rho_v(r) = m\langle \mathcal{N}(r)/\mathcal{V}(r)\rangle$, where $\langle \cdots \rangle$ indicates an average over the configurations sampled in our simulations. As shown in Fig. 4a and Supplementary Fig. 1, the oscillations observed in $\rho_o(r)$ are absent in $\rho_v(r)$, allowing more precise tracking of the density variation in the droplet interior. Note that the Voronoi cells for molecules at the droplet surface have a divergent volume, and so $\rho_v(r)$ vanishes for the outer-most molecular layer.

Figure 4b and Supplementary Fig. 2 show $\rho_v(r)$ for a wide range of $N$ and $T$, and reveal complex changes in internal structure. In particular, we observe the emergence of a density maximum as $N$ increases. The density at all $r$ for our smallest droplets ($N = 100$)

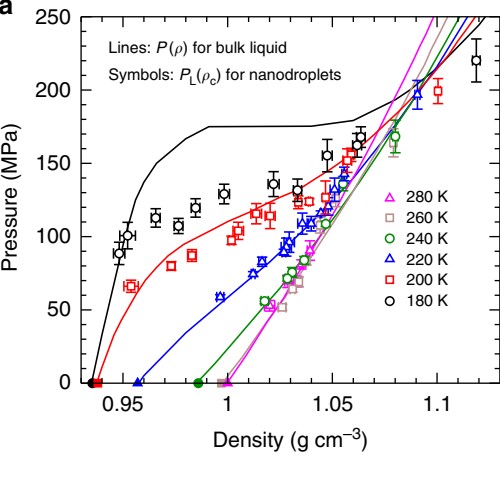

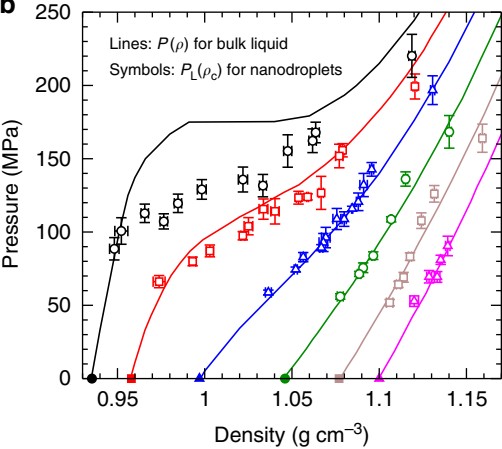

**Fig. 3** Equations of state for bulk liquid and nanodroplets of TIP4P/2005. **a** Isotherms of $P(\rho)$ for the bulk liquid (solid lines), taken from the EOS presented in ref. [21]; and $P_L(\rho_c)$ for water nanodroplets (open symbols). $N$ decreases with increasing density along each isotherm. The filled symbols locate the bulk behaviour expected for droplets as $R \to \infty$ and $N \to \infty$. Lines and symbols of the same colour correspond to the same $T$. **b** Same as in **a**, but to permit easier examination of each isotherm, all data for $T = 200$ K have been shifted horizontally by 0.02 g cm$^{-3}$; for 220 K by 0.04 g cm$^{-3}$; for 240 K by 0.06 g cm$^{-3}$; for 260 K by 0.08 g cm$^{-3}$; and for 280 K by 0.10 g cm$^{-3}$. Legend is the same as in **a**. In both panels, error bars represent one standard deviation of the mean.

increases monotonically as $T$ decreases. For $N = 360$, the density near the centre passes through a maximum as $T$ decreases, although the surface density still increases monotonically. For larger droplets (e.g., $N = 776$), the density at almost all $r$ passes through a maximum as $T$ decreases.

We define the droplet core density as $\rho_c = m\langle \mathcal{N}_c/\mathcal{V}_c\rangle$, where $\mathcal{N}_c$ is the number of O atoms within $r_c = 0.5$ nm of the droplet centre, and $\mathcal{V}_c$ is the total volume of the Voronoi cells for these atoms. (For $N \leq 205$ we use $r_c = 0.25$ nm, since for our smallest droplets the effect of the surface extends closer to the centre.) Fig. 5 shows $\rho_c$ as a function of $T$ for fixed $N$, and confirms that a density maximum occurs in the core of water nanodroplets as small as $N = 360$.

The density maximum of bulk water occurs as its random tetrahedral network (RTN) structure becomes more prominent as $T$ decreases[12]. At low $T$, we find that $\rho_c$ tends towards the density of the bulk RTN (~0.94 g cm$^{-3}$) for our larger nanodroplets. Despite the disruption of bulk-like structure occurring at the nanodroplet surface, the evolution of our density profiles as $T$

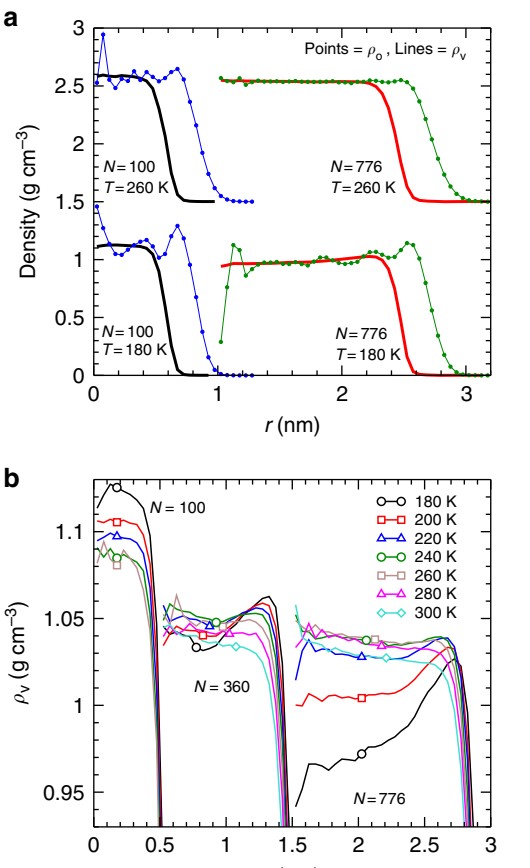

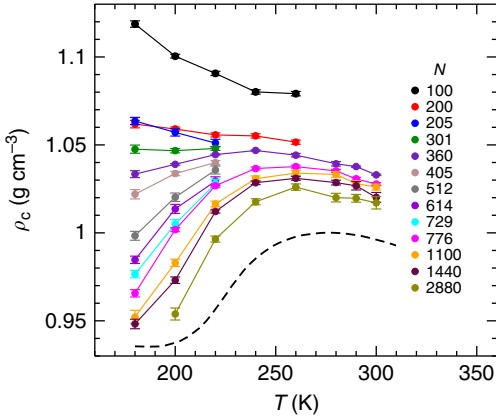

**Fig. 5** Density maximum of liquid nanodroplets. Plot of $\rho_c$ versus $T$ for water nanodroplets of fixed $N$ (symbols). The dashed line is the $P = 0$ isobar of $\rho$ for bulk TIP4P/2005 water taken from ref. [21], which corresponds to the expected behaviour of $\rho_c$ for droplets as $N \to \infty$. Error bars represent one standard deviation of the mean.

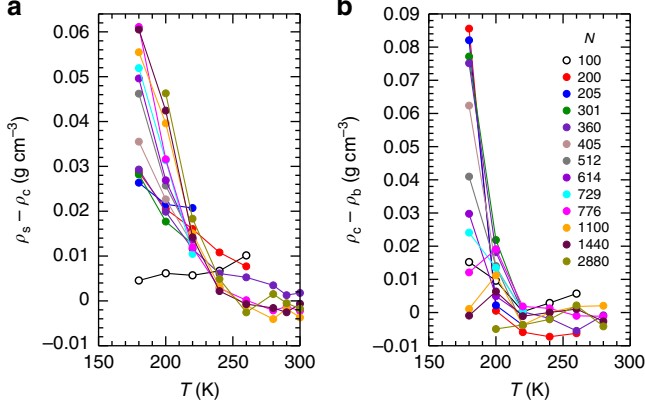

**Fig. 6** Emergence of density differences within nanodroplets with decreasing temperature. **a** $\rho_s - \rho_c$ versus $T$, and **b** $\rho_c - \rho_b$ versus $T$, for nanodroplets of various $N$, as indicated in the legend.

**Fig. 4** Nanodroplet density profiles. **a** Density profiles $\rho_o(r)$ (symbols) and $\rho_v(r)$ (lines) for water nanodroplets at various $N$ and $T$. For $T = 260$ K, the curves have been shifted vertically by 1.5 g cm$^{-3}$. For $N = 776$, the curves have been shifted horizontally by 1 nm. **b** $\rho_v(r)$ at various $N$ and $T$. For $N = 360$, the curves have been shifted horizontally by 0.5 nm; and for $N = 776$ by 1.5 nm. In **a** and **b**, note that the error increases as $r \to 0$; see Supplementary Figs. 1 and 2.

decreases is thus driven by the formation of a low-density RTN in the droplet core. A similar low density core was observed in recent simulations of glassy water nanoparticles[29]. Notably, the onset of ice crystallisation in nanodroplets is observed experimentally also when $N$ reaches 250–300[24], consistent with our finding that this is the range of $N$ in which a density maximum and a RTN structure emerge with decreasing $T$.

Our results also show that, as a consequence of RTN formation in the droplet core, the density profile of a nanoscale water droplet undergoes a dramatic 'density inversion' as $T$ decreases: As shown in Fig. 4b and Supplementary Fig. 2, high-$T$ droplets have a denser core and a slightly less dense surface, as expected for a simple liquid droplet, while low-$T$ droplets have a less dense core and a distinctly denser surface. In Methods, we describe a procedure to define the maximum density $\rho_s$ in the surface region of $\rho_v(r)$. In Fig. 6a we plot the difference $\rho_s - \rho_c$ as a function of $T$. We find for all $N \geq 200$ that $\rho_s - \rho_c$ is slightly negative at high $T$ and is positive and rapidly increasing at low $T$. Despite the emergence of the RTN in the droplet core as $T$ decreases, the equilibrium droplet structure at low $T$ always exhibits a higher density liquid layer at the interface with the vapour.

**Laplace pressure and equation of state for nanodroplets.** At low $T$, $\rho_c$ varies by >15%, suggesting that $P_L$ inside our droplets changes considerably with $N$. To measure $P_L$ directly, rather than

relying on the Young–Laplace equation, we evaluate the configurational contributions to the tangential and normal components of the pressure, $P_T$ and $P_N$, as functions of $r$, shown in Fig. 7a and Supplementary Fig. 3[20, 31]. We find that there is a region within each droplet where $P_T \simeq P_N$, as occurs in a bulk liquid, and we define $P_L$ as the average of the total pressure $P_{tot}$ in this region (see Methods). In Fig. 7b we see that isotherms of $P_L$ are proportional to $R^{-1}$, consistent with the prediction of the Young–Laplace equation. Figure 7b confirms that the variation of $P_L$ with $N$ is large, reaching >200 MPa for our smallest nanodroplets at low $T$.

We estimate $\gamma$ from the slopes of the isotherms in Fig. 7b. In Supplementary Fig. 4 we compare our $\gamma$ values to results obtained previously using TIP4P/2005 for the surface tension $\gamma_p$ of a planar liquid–vapour interface[32]. Although the $T$ ranges of the two data sets do not overlap, our result at 280 K is quantitatively consistent with the value of $\gamma_p$ at 300 K. This agreement, and the linearity of the isotherms in Fig. 7b, suggests that $\gamma$ for a strongly curved interface (our results) and a flat one ($\gamma_p$ from ref. [32]) differ little, i.e., that the Tolman length may be close to zero[33]. On the other hand, our results for $\gamma$ increase more rapidly with decreasing $T$ than indicated by the low-$T$ extrapolation of $\gamma_p$ given in ref. [32]. This difference may arise due to the emergence at low $T$ of the complex and inverted density profiles shown in Fig. 4b, or of

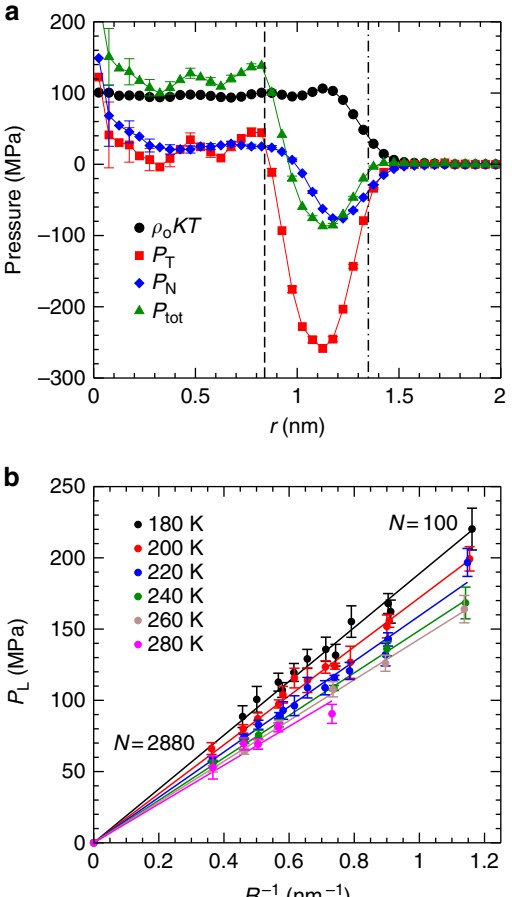

**Fig. 7** Laplace pressure inside nanodroplets. **a** Contributions to the pressure inside a water nanodroplet as a function of $r$, for $N = 360$ and $T = 200$ K. Vertical lines identify $r = R_L$ (dashed) and $r = R$ (dot-dashed). **b** Isotherms of $P_L$ as a function of $R^{-1}$. Note that $N$ decreases with $R^{-1}$ along each isotherm. The point at the origin is the expected value of $P_L$ for droplets as $R \to \infty$. The straight lines are fits of the Young-Laplace equation $P_L = 2\gamma/R$ to the data along each isotherm. The error for $R^{-1}$ is smaller than the symbol size; the error in $P_L$ represents one standard deviation of the mean.

curvature effects, or both. Further work is required to clarify these influences.

We compare in Fig. 3 the correspondence between the EOS of the bulk liquid, and the variation of $P_L$ with $\rho_c$ along isotherms in our nanodroplets. For $T \geq 220$ K, we find that the bulk and nanodroplet EOS isotherms agree within statistical error for all $N$. Our results thus show that the density maximum observed in Fig. 5, which occurs in the range 240–260 K, is a consequence of the ability of nanodroplets to follow the bulk EOS for $T \geq 220$ K, where the bulk density maximum also occurs. Interestingly, we also find that the absence of a density maximum at small $N$ in Fig. 5 is not due to deviations from the bulk EOS. Instead, the density maximum disappears because the path followed by a nanodroplet of fixed $N$ in the EOS deviates strongly from an isobar for small $N$, as shown in Fig. 8a.

**Nanodroplet behaviour approaching the LLPT.** Despite the good correspondence in Fig. 3 between the nanodroplet and bulk EOS for $T \geq 220$ K, we find that the agreement breaks down for $T \leq 200$ K. Specifically, the data points on the nanodroplet isotherms for 200 and 180 K lie at higher density than the bulk for all but our largest droplets. We quantify this deviation in Fig. 6b, where we plot the difference between $\rho_c$ for a given droplet, and

$\rho_b$, the density of a bulk liquid having the same $T$ and $P = P_L$ as the droplet. Not surprisingly, the largest droplets maintain their bulk-like properties at all $T$, as they must in the limit $N \to \infty$. However, for $N \leq 776$, we observe a dramatic loss of bulk-like character at low $T$. An interesting exception to this trend occurs for our smallest droplet ($N = 100$), which shows only a modest deviation compared e.g., to $N = 200$.

This complex behaviour can be understood by considering the influence of the LLPT that occurs in TIP4P/2005 on the shape of the bulk EOS (Fig. 3), in concert with the unusual density profiles observed in our droplets (Fig. 4). Figure 8a shows our nanodroplet EOS data plotted so as to highlight the path in the $\rho_c - P_L$ plane followed by a droplet of fixed $N$ as $T$ decreases. For all droplets with $N \geq 200$, we find that $P_L$ is less than $P_c$ for the LLPT of TIP4P/2005. On cooling, $\rho_c$ for these droplets tends towards the low density liquid (LDL) branch of the bulk EOS. These droplets also develop inverted density profiles as shown in Fig. 4, in which $\rho_c$ separates from $\rho_s$ at low $T$; this growing separation is illustrated in the density-pressure plane in Fig. 8b. Since the surface remains dense, a large density gradient must occur in order for $\rho_c$ to reach $\rho_b$ at low $T$. Although our largest droplets are big enough to accommodate the required gradient, for $N \leq 776$ we find that the droplets are too small for $\rho_c$ to reach $\rho_b$ (see Supplementary Fig. 5). As a consequence, bulk-like properties are not attained in the cores of our smaller droplets ($200 \leq N \leq 776$) at low $T$, resulting in the EOS deviations observed in Fig. 3.

In the case of the $N = 100$ droplet, $P_L$ exceeds $P_c$ at low $T$, and the droplet enters the region of the bulk EOS associated with the high density liquid (HDL); see Fig. 8. For the $N = 100$ droplet, $\rho_c$ is comparable to $\rho_s$, and both are close to the bulk HDL density (Fig. 8b). The signature of an inverted density profile is also weak for $N = 100$ (Fig. 6a). For $P_L > P_c$, we thus find that the droplet behaviour changes suddenly to that of a simple liquid. In sum, our results demonstrate that the droplet density profile correlates well to the bulk regime of the LLPT explored by the droplet: As $T \to T_c$, an inverted density profile indicates a droplet for which $P_L < P_c$, while a monotonic density profile suggests that $P_L > P_c$.

The large radial density change observed in our droplets at low $T$ is perhaps suggestive of nanoscale phase separation in which a LDL-like core is wetted by a HDL-like surface layer. Figure 8b shows that $\rho_c$ and $\rho_s$ grow farther apart as $T$ decreases for all droplets having $P_L < P_c$. The values of $\rho_s$ are consistent with the range expected for HDL, while $\rho_c$ is lower, and approaches LDL-like values when the droplet is large enough. Certainly, intrinsic surface effects play a large role in determining our density profiles: A droplet with a low-density RTN in the core will have a disrupted RTN, and therefore higher density, near the interface with the vapour. Independent of surface effects, the LLPT of TIP4P/2005 also promotes the appearance of distinct high and low density regions near the critical conditions. These two effects are mutually reinforcing, and it is likely that both contribute to the large density variations in our low $T$ nanodroplets.

A bulk response function such as $K_T$, which quantifies volume fluctuations, diverges at the critical point of a LLPT. To test for a similar effect in our nanodroplets, we use the fluctuations of the Voronoi volumes for a subsystem of molecules inside our droplet cores to define a quantity $K_T^s$ which is analogous to $K_T$ (see Methods). As shown in Fig. 9, we observe a growing maximum in $K_T^s$ along isotherms at $T = 200$ and 180 K, the same $T$ for which strong deviations emerge between the nanodroplet and bulk EOS. This behaviour confirms that the interiors of our coldest droplets exhibit effects directly associated with the approaching LLPT in TIP4P/2005. Experiments have recently provided strong evidence of a $K_T$ maximum in supercooled water, both for water enclosed in micrometre quartz inclusions[34], and for unsupported

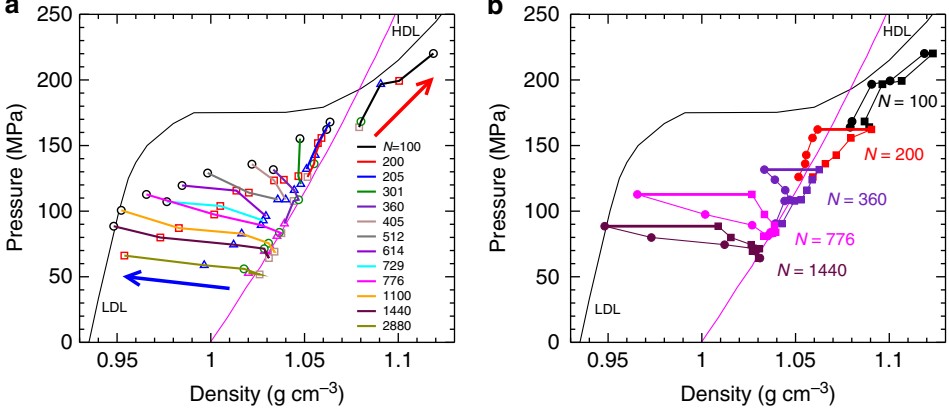

**Fig. 8** Variation of density and pressure with temperature inside nanodroplets of various sizes. **a** Same data as in Fig. 3a for $P_L$ versus $\rho_c$, except here data points with the same $N$ are connected by coloured lines as indicated in the legend. Symbol types have the same meaning as in Fig. 3a. Error bars are omitted for clarity. The blue and red arrows indicate the direction of decreasing $T$ along a curve of constant $N$. Note that for large $N$, lines of constant $N$ are nearly isobaric, whereas as $N$ becomes small, strong deviations from isobaric behaviour are observed. **b** $P_L$ versus $\rho_c$ (circles), and $P_L$ versus $\rho_s$ (squares) for droplets of various $N$. Note that $T$ varies along each curve, from 280 to 180 K (for $N = 360$, 776, and 1440) and from 260 to 180 K (for $N = 200$ and 100). Data points for $\rho_c$ and $\rho_s$ at 180 K with the same $N$ are joined by a thick horizontal line, to highlight their difference at low $T$. In both **a** and **b**, isotherms of $P$ versus $\rho$ for the bulk liquid are shown for $T = 180$ K (thin black line) and 280 K (thin magenta line). The branches of the 180 K bulk isotherm corresponding to LDL and HDL are indicated.

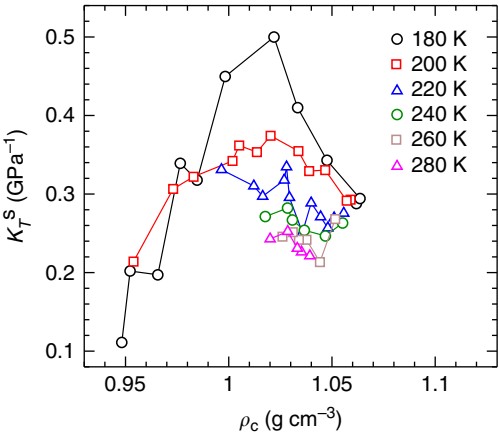

**Fig. 9** Isothermal compressibility of the nanodroplet core. Isotherms of $K_T^s$ versus $\rho_c$ for $200 \leq N \leq 2880$.

microdroplets[17]. Our results show that this effect may also be observable in much smaller nanodroplets, which allow even deeper supercooling, and which access higher $P$ closer to the estimated critical conditions of the LLPT proposed for real water.

## Discussion

It is a central goal of nanoscience to determine the scale at which macroscopic behaviour first emerges. Our results show that bulk-like liquid properties, including the density maximum, can be observed using water nanodroplets as small as several hundred molecules. We also demonstrate that by varying $N$, the interiors of water nanodroplets explore a remarkably wide range of both density and pressure. This range is large enough to encompass and to reproduce the pattern of thermodynamic anomalies that occurs in bulk water for $T \geq 220$ K, including precursors of the proposed LLPT. Indeed, we have shown that simply measuring the nanodroplet size $R$ as a function of $N$ and $T$ is a viable approach for revealing the qualitative signatures of these anomalies.

For $T \leq 200$ K, we observe dramatic departures from bulk-like behaviour, which arise as $T$ approaches $T_c$ for the LLPT of TIP4P/2005. It is well understood that the discontinuities at a bulk phase transition are strongly rounded and shifted by finite-size effects in small systems[35]. Explicit surface effects in nanoscale systems also induce deviations from bulk behaviour. These two effects are intertwined in our system, and together they generate the complex evolution of nanodroplet properties that we observe as $T$ decreases. Disentangling the relative contributions of finite-size and surface effects is challenging. For example, consider the $\rho_c$ data shown in Fig. 5. In the bulk liquid, isobars of the density will decrease more sharply with decreasing $T$ as $P \to P_c$. Although our smaller nanodroplets access higher $P$, the variation of $\rho_c$ does not sharpen. Finite-size effects are at least partially responsible, since we know that the phase transition exists in our model bulk system, but we also know that $\rho_c$ does not reach the bulk value at low $T$ and small $N$ because of the influence of the dense surface layer, as discussed above. Further work to quantify how finite-size and surface effects combine to produce the novel phenomena observed here would be valuable, for example, to better understand the unusual shape of our density profiles at low $T$.

Despite these complexities, our results establish the pattern of nanodroplet behaviour that occurs in a water-like system that exhibits a bulk LLPT. The key features of this behaviour are the deviation of nanodroplet properties from the bulk as $T \to T_c$, and the emergence of a large and inverted gradient in the droplet density profile when $P_L < P_c$. These observations have the potential to assist in understanding many systems where water nanodroplets play a central role. For example, for aerosols involved in cloud formation[2], the average position and chemical activity of a solute molecule within a water nanodroplet may be strongly influenced by the changes in the pressure and the density profile that we observe on cooling[36, 37]. Regarding the ongoing efforts to clarify the behaviour of deeply supercooled water, experiments increasingly exploit small water droplets, from the microscale[16–18] to the nanoscale[7, 11, 28]. We confirm here that cold water nanodroplets both resist crystallisation and generate sufficient Laplace pressure to directly access the region of the proposed LLPT. Furthermore, our results suggest specific ways to use nanodroplets to help locate a possible LLPT. For example, an experimental probe sensitive to the droplet density profile, or to

the volume fluctuations of the droplet core, could identify droplets that have entered the critical regime, from which an estimate of $T_c$ and $P_c$ might be obtained.

## Methods

**Computer simulations.** We simulate liquid water nanodroplets of size $N =$ 100–2880 molecules. The molecules interact via the TIP4P/2005 water pair potential[30]. We use Gromacs v4.6.1[38] to carry out our molecular dynamics (MD) simulations. The equations of motion are integrated using the leap-frog algorithm, with a time step of 2 fs. We carry out simulations in the fixed-$(N, V, T)$ ensemble, where $N$ is the total number of molecules in the simulation cell, and $V$ is the volume of the cell. We hold the temperature constant using a Nosé–Hoover thermostat with a time constant of 0.1 ps. Our droplets are located in a cubic simulation cell, with periodic boundary conditions, of various sizes $V = L^3$, as listed in Supplementary Tables I–III. The intermolecular interaction is set to zero for molecules separated by more than $L/2$. We choose $L$ large enough relative to the size of the droplet to avoid any direct interaction between the water droplet and its periodic images. Consequently, all molecules within a nanodroplet interact directly, without cut-offs or approximations for long-range electrostatic interactions.

Individual molecules occasionally escape from the surface of the droplet and contribute to a vapour phase in equilibrium with the droplet. We choose $L$ small enough so that the average number of molecules in the vapour phase is never >0.004$N$ (see Supplementary Tables I–III). The vapour pressure in our simulations is always much smaller than the size of the error in our estimates for $P_L$, and so we consider the vapour pressure to be zero. We note that because of the presence of the vapour phase, and our droplet definition (see below), the average number of molecules in a nanodroplet $N_d$ may differ from the number of molecules in the system $N$. However, as stated above the difference is always <0.004$N$, and for $T \leq$ 220 K we find no difference. Having distinguished here between the definitions of $N$ and $N_d$, we note that to calculate the values of $\rho_R$ presented in Fig. 2, we use $\rho_R = 3mN_d/4\pi R^3$. When labelling data in our figures, we use the $N$ value for the run from which the data are obtained.

For $N = 1440$ and 2880, we create initial configurations by placing $N$ molecules at random within the simulation cell, and then running for long enough so that the molecules condense into a single droplet. Initial configurations for other values of $N$ are obtained from our $N = 1440$ configurations by deleting molecules from the surface until the desired $N$ is reached.

We conduct two types of run to evaluate the equilibrium properties of our droplets: conventional 'single long runs' (SLR), and 'swarm relaxation' runs[39]. We use SLRs for droplet sizes $N = 100, 200, 360, 776, 1100, 1440$, and 2880; see Supplementary Tables I–II. In each SLR, the system comes into equilibrium during the first phase of the run, followed by a production phase from which equilibrium configurations are harvested. The relaxation time $\tau$ (defined below) is evaluated from the production phase. In each SLR, our equilibration phase is at least 10$\tau$ long, and is never <100 ns. The length of the production phase of our runs is never less than 46$\tau$, and is typically between $10^2\tau$ and $10^3\tau$.

For droplet sizes $N = 205, 301, 405, 512, 614$, and 729 we use the 'swarm relaxation' method, described in detail in ref. [39]. The initial configurations for these choices of $N$ are obtained from a SLR configuration for $N = 2880$ by deleting molecules from the surface until the desired $N$ is reached. We then run each new configuration for 350 ns at 200 K to generate a starting configuration for our swarm relaxation runs. For our swarm relaxation runs at 220 K, $M$ different initial configurations are generated by randomising the velocities of the starting configuration according to a Maxwell–Boltzmann distribution appropriate for $T =$ 220 K. We use $M = 250$ or 1000, as documented in Supplementary Table III. We then conduct an ensemble of $M$ independent runs (a 'swarm'), and monitor the average behaviour of the swarm over time to determine when the runs have attained equilibrium. Swarm relaxation runs at 200 K (180 K) are initiated using the $M$ final configurations from the 220 K (200 K) runs. We evaluate the relaxation time $\tau_s$ of each swarm ensemble from the autocorrelation function of the system potential energy. As shown in ref. [39], swarm runs of length 10$\tau_s$ are sufficient for reaching equilibrium. Supplementary Table III shows that the run time $t_{run}$ for each of our swarm runs significantly exceeds this threshold. To estimate equilibrium properties, we carry out an ensemble average over the final $M$ configurations of each swarm run.

**Droplet definition.** We define the droplet as the largest cluster of water molecules in our system. A molecule belongs to a cluster if its distance to any molecule in the cluster is less than 0.35 nm[40].

**Relaxation times.** To determine the structural relaxation time $\tau$ of the droplets in our SLRs, we use the method of refs [41, 42]. We evaluate the bond correlation function $\phi(t)$, which characterises the likelihood that a bond present at time $t = 0$ remains unbroken at time $t$:

$$\phi(t) = \left\langle \frac{1}{N_B(0)} \sum_{i<j} n_{ij}(t)\, n_{ij}(0) \right\rangle. \qquad (1)$$

Here, $n_{ij}(t) = 1$ for all $t$ up to the time that the bond between molecules $i$ and $j$ breaks for the first time. After the bond breaks, $n_{ij}(t) = 0$ for all time, even if the bond later reforms. Molecules $i$ and $j$ are considered bonded if the distance between their O atoms $r_{ij} \leq 0.32$ nm. $N_B(0)$ is the number of bonds at $t = 0$. The average in Eq. 1 is taken over multiple choices of the time origin $t = 0$.

In all cases, we find that $\phi(t)$ decays to zero on a time scale much shorter than the length our SLRs. This behaviour confirms that all of our nanodroplets are equilibrium liquid droplets, and not glassy solids. We define $\tau$ as the time such that $\phi(\tau) = e^{-1}$. We define the number of independent configurations in each of our SLR simulations as $N_\tau = t_{run}/\tau$, where $t_{run}$ is the total length of the production phase of a SLR. The values of $\tau$ and $N_\tau$ for each of our SLRs are listed in Supplementary Tables I–II.

**Testing for crystal formation.** To determine if crystalline ice forms in our liquid nanodroplets, we use the procedure developed by Frenkel and coworkers[43, 44] to identify clusters of crystal-like molecules, based on quantifying the local bond order using spherical harmonics[45]. The specific procedure we use to identify ice-like clusters is the same as that described in ref. [46]. We monitor $n_{max}$, the size of the largest ice-like cluster in the droplet, as a function of time during our SLR simulations. The largest value of $n_{max}$ encountered in all of our SLR simulations is 12, observed in our $N = 1100$ droplet at 180 K. In the same run, the average value of $n_{max}$ is 1.4. All such ice-like clusters appear only as transients, and dissipate on a time scale comparable to $\tau$. These observations confirm that our droplets remain in the liquid phase on the time scale of our simulations.

**Stability of liquid nanodroplets at low $T$.** The coexistence temperature for the bulk liquid and ice Ih phases of TIP4P/2005 is 252 K at ambient $P$, and decreases to 230 K at $P = 200$ MPa[30]. To estimate the minimum $T$ at which we observe a thermodynamically stable liquid droplet, we prepare approximately spherical nanocrystallites of ice Ih of size $N = 360$ and 776. We run each of these nano-crystallites for 4 ns at $T = 180, 200, 220, 240$, and 260 K. During each run we monitor $n_{max}$, the size of the largest crystalline cluster as a function of time $t$, using the definition of $n_{max}$ described in ref. [46]. As shown in Supplementary Fig. 6, our $N = 360$ system completely melts within 4 ns for $T \geq 200$ K, and our $N = 776$ system melts within 4 ns for $T \geq 220$ K. This behaviour demonstrates that liquid nanodroplets of these sizes are thermodynamically stable below the melting temperature for the bulk liquid phase.

**Droplet radius.** To quantify the droplet radius, we model the droplet as an ellipsoid with uniform density[47, 48]. We first compute the moment of inertia tensor $I$ from the position vector $\mathbf{r}_i$ for the centre of mass of each molecule $i$ in the droplet, relative to the droplet centre of mass. The elements of $I$ are given by,

$$I_{jk} = m \sum_{i=1}^{N_d} \left( r_i^2\, \delta_{jk} - r_{ij}\, r_{ik} \right) \qquad (2)$$

where $r_i = |\mathbf{r}_i|$; $r_{ij}$ is the $j^{th}$ component ($x$, $y$ or $z$) of $\mathbf{r}_i$; and $\delta_{jk}$ is the Kronecker delta. The eigenvalues of $I$ ($I_{xx}$, $I_{yy}$, and $I_{zz}$) are related to the lengths of the principal axes ($a$, $b$ and $c$) of the ellipsoid via the relations: $5I_{zz} = mN_d(a^2 + b^2)$; $5I_{xx} = mN_d(b^2 + c^2)$; and $5I_{yy} = mN_d(a^2 + c^2)$. We then define the droplet radius as $R = (abc)^{1/3}$. The values of $R$ reported here are averages over the ensemble of droplet configurations generated for each $N$ and $T$. We note that the qualitative pattern of behaviour observed in Fig. 2 does not change if we define $R$ instead as the radius of gyration.

**Voronoi volumes and isothermal compressibility.** To evaluate the volumes of the Voronoi cells around the O atoms in our nanodroplet configurations, we use the 'Voro++' software described in ref. [49].

To define a quantity similar to $K_T$ in our droplet cores, we exploit the dependence of $K_T$ on the volume fluctuations in a fixed-$(N, P, T)$ ensemble: $K_T = \langle \delta V^2 \rangle / \langle V \rangle kT$, where $\langle \delta V^2 \rangle$ is the variance of the system volume $V$[50]. We define a fixed-$(N, P, T)$ subsystem within the droplet core by selecting from each configuration the 40 molecules that are closest to the droplet centre of mass. We choose 40 molecules because this is approximately the number of molecules within the core region of our $N = 200$ droplets, allowing us to consider a subsystem of fixed size throughout the range $200 \leq N \leq 2880$. We define the volume of the subsystem $\mathcal{V}_s$ as the sum of the Voronoi volumes of these 40 molecules, and thereby define $K_T^s = \langle \delta \mathcal{V}_s^2 \rangle / \langle \mathcal{V}_s \rangle kT$. Figure 9 plots our results for $K_T^s$ along isotherms for $N \geq 200$ as a function of $\rho_c$.

**Surface region of density profiles.** To define the portion of $\rho_v(r)$ associated with the droplet surface, we first model our data for $\rho_v(r)$ by fitting to,

$$\rho_{fit}(r) = \frac{\rho_0}{2}\left[ \tanh\left( \frac{r - r_0}{\sigma_0} \right) + 1 \right], \qquad (3)$$

where $\rho_0$, $r_0$ and $\sigma_0$ are fit parameters. To conduct this fit, we ignore data for $r < 0.2$ nm, to avoid the larger error in $\rho_v(r)$ at small $r$; see Supplementary Fig. 2. We define the surface region of $\rho_v(r)$ as the region $r > r_0 - 0.6$ nm, and $\rho_s$ as the maximum

value of $\rho_v(r)$ in the surface region. The density difference between the droplet surface and the core, $\rho_s - \rho_c$, is plotted in Fig. 6a.

**Laplace pressure**. To find the Laplace pressure $P_L$, we first evaluate $P_N$ and $P_T$, the normal and tangential components of the configurational contribution to the pressure as a function of $r$ within our droplets. We use the approach presented in ref. [31], modified to suit the case of a rigid molecular model of water as described in ref. [20]. As illustrated in Fig. 7a and Supplementary Fig. 3, we find in all cases that $P_N$ and $P_T$ differ, and display a prominent minimum, near the droplet surface. For smaller $r$, $P_N$ and $P_T$ become approximately equal within the error of our calculations. In a bulk liquid, the pressure tensor is isotropic, and so we identify the region inside the droplet where $P_N \cong P_T$ as a bulk-like region in which the average total pressure is the Laplace pressure $P_L$. The total pressure is given by,

$$P_{tot} = \frac{1}{3}P_N + \frac{2}{3}P_T + \rho_o kT, \tag{4}$$

where $k$ is Boltzmann's constant. To evaluate $P_L$, we average $P_{tot}$ from $r = 0$ to $r = R_L$, where $R_L$ is the radius at which $P_N$ and $P_T$ first cross as $r$ decreases below the surface region where the minima in $P_N$ and $P_T$ occur.

**Error estimates**. All error bars presented in our figures represent $\pm \sigma / \sqrt{N_s}$, where $\sigma$ is the standard deviation of the measured quantity, and $N_s$ is the number of independent configurations averaged over. For our SLRs, we use $N_s = N_\tau$. For our swarm runs, we use $N_s = M$.

**Data availability**. The data that support the findings of this study are available from the authors on reasonable request.

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

## Acknowledgements

I.S.-V. and P.H.P. thank NSERC for support. Computational resources were provided by ACENET and Compute Canada. We thank R.K. Bowles and F. Sciortino for helpful discussions.

## Author contributions

All authors contributed equally to planning the study, interpreting the results, and preparing the manuscript. S.M.A.M. carried out all simulations and data analysis.

## Additional information

**Competing interests:** The authors declare no competing interests.

