## [Peer Review File · Nature Communications]

Reviewer #1 (Remarks to the Author):

This paper addresses the question if water anomalies that are typically found in bulk liquid water can be observed in nanodroplets of varying size using TIP4P/2005 simulations. The authors nicely demonstrate that the radius of droplets correspond to different pressure and follows similar trends in the equation of state (EOS) for bulk water. The paper further points to that experimental investigations of a critical point where higher pressures need to be reached during fast cooling could be accomplished with nanodroplets of varying size. It is also a significant with density variations within the droplets at certain conditions pointing to something that could be a trace of a phase transition and co-existence. I find this paper to be interest for the community and recommends publication after some comments have been addressed.

1. In the EOS is it possible to extract if compressibility is diverging towards something similar to a Widom line or a LLC. It could be valuable with a plot of compressibility versus T for different sizes and to compare this with bulk water with the corresponding pressures.
2. If there is a precursor to a LLC under certain conditions would we not expect that fig. 5 shows sharper transition than for bulk water at 1bar. Some discussion of this would be valuable.
3. Why is there a lower density region inside the droplets under certain condition and not on the surface facing the vacuum. It would be good with some discussion on this point.
4. In the introduction it is indicated that the freezing temperature decreases for nanodroplets. This could be misleading since what matters is not the freezing temperature but timescales since we are in a metastable region. It is known that nanodroplets have a higher nucleation rate than bulk water and the reason for seeing crystallization at lower temperatures in nanodroplets is related to the much faster cooling rates.

Reviewer #2 (Remarks to the Author):

Report on « Thermodynamic anomalies of water nanodroplets »

This paper reports on an extensive simulation study with the TIP4P/2005 model for water of the thermodynamic properties of cold nanodroplets. Cold water nanodroplets occur in Nature and have been studied experimentally. An outstanding question is how their properties are related to that of bulk water. The present study convincingly demonstrates that the bulk equation of state is very well

followed even for very small droplets. This result is very significant as it could motivate further experimental studies to access a poorly known region of the phase diagram of water. The main physical effect of the small curvature, the Laplace pressure, is well studied and understood, and this will also be of interest for physicists studying how macroscopic laws transpose to the nanoscale.

The article is very clearly written; all technical details are given and will allow other researchers to reproduce the study. I recommend publication. I have only very few suggestions to slightly improve the context, which the authors may choose to follow or not.

1) In the introduction, the authors give $T_h=232$ K. T_h is an experiment-dependent quantity, see for instance F. Caupin, *J. Non-Cryst. Solids* 407 (2015) 441–448 for a discussion. Indeed, Ref. 13 reports T_h around 227 K for microdroplets, which may be considered bulk. Note however that a recent work on microdroplets evaporating in vacuum, which measured their radii and deduced their temperature [C. Goy et al. to appear in *Phys. Rev. Lett.*

https://journals.aps.org/prl/accepted/d5076Y88R5415d6e36c88093531_548bd954946eb2 available on arxiv

<https://arxiv.org/abs/1711.02412>

] shows that the temperature reported in Ref. 13 might be underestimated, and that the actual temperature might rather be around 230 K.

2) The authors write “experimental efforts increasingly exploit water nanodroplets to probe deeply supercooled conditions for bulk water” and cite Refs. 7 and 13. Ref. 7 does study nanodroplets with radii between 3.2 and 5.8 nm, but Ref. 13 rather studies microdroplets, with radii ranging from 4.5 to 18.5 μm . Other references from the reference list would be more appropriate. A pioneering work on water nanodroplets could also be added: J. Huang, L.S. Bartell, Kinetics of homogeneous nucleation in the freezing of large water clusters, *J. Phys. Chem.* 99 (1995) 3924–3931.

3) Although I understand this is not the focus of the present work, it would be interesting if the authors could compare the values they find for the surface tension and the thickness of the liquid-vapor interface with literature values. They just write “ γ is increasing with decreasing T , consistent with previous work [25]”. Are the values consistent? Is there any effect of curvature (Tolman correction)? See F. Caupin *Phys. Rev. E* 71, 051605 (2005) and references therein about the thickness of the liquid-vapor interface, and N. Bruot and F. Caupin, *Phys. Rev. Lett.* 116, 056102 (2016) and references therein about the effect of curvature.

Reviewer #3 (Remarks to the Author):

In this work, the authors calculated the density and internal pressure of water nanodroplets with different sizes and temperatures by using TIP4P/2005 water model. The results show a density maximum, which is one of the anomalous behaviors of bulk water, also exists in the central region of water nanodroplets as small as containing a few hundred water molecules. The internal pressure of these droplets, resulted from Laplace pressure, could also be very high for very small droplets. The authors thus argue that the signature of thermodynamic anomalies of bulk water including the precursor of debated liquid-liquid phase transition may also occur in water nanodroplets.

Although this is no doubt an interesting work carried out with great care, I feel the main results/conclusions are quite not unexpected. It is true that water nanodroplets are considered different from bulk water due to both significant surface/volume ratio and large internal pressure, but beneath the molecular layer of those droplets, there exists a “bulk-like” region in these droplets which behaves like bulk water with a pressure comparable to the corresponding Laplace pressure of droplet. This has been indicated by some previous studies. This work certainly provides a more detailed, rigorous analysis to prove this point, but the main conclusion does not appear as a surprise, but rather reinforcement of what has been speculated. On the other hand, the significance of very small nanodropets, particularly in hope of resolving the much-debated LLPT, is perhaps also limited. This can be seen by the divergence of EOS between droplets and bulk water at low temperature (Fig. 3). As noted by authors, finite size effect can significantly suppress the density discontinuity near critical point. Overall I think the scope of the paper may be more suitable for a more specialized journal.

A few minor comments:

1. In both abstract and main text, the authors claim a density maximum occurs in a droplet as small as $N \sim 300$. This is not totally clear on Fig. 5, as there are only three data points and a maximum is not evident yet. Instead this is clear for $N=360$.
2. What is the error bar for local density in Fig. 4b?

Reviewer #1

This paper addresses the question if water anomalies that are typically found in bulk liquid water can be observed in nanodroplets of varying size using TI4P/2005 simulations. The authors nicely demonstrate that the radius of droplets correspond to different pressure and follows similar trends in the equation of state (EOS) for bulk water. The paper further points to that experimental investigations of a critical point where higher pressures need to be reached during fast cooling could be accomplished with nanodroplets of varying size. It is also a significant with density variations within the droplets at certain conditions pointing to something that could be a trace of a phase transition and co-existence. I find this paper to be interest for the community and recommends publication after some comments have been addressed.

1. In the EOS is it possible to extract if compressibility is diverging towards something similar to a Widom line or a LLCPC. It could be valuable with a plot of compressibility versus T for different sizes and to compare this with bulk water with the corresponding pressures.

This suggestion motivated us to seek a way to directly calculate a quantity similar to the bulk isothermal compressibility K_T in our droplets. As described in Methods, we are able to do so by measuring the fluctuations of the Voronoi volumes for a subsystem of molecules near our droplet centres, and we denote this new measure of fluctuations as K_T^s . These new results are presented in Fig. 9, as isotherms of K_T^s versus ρ . We observe a prominent maximum in K_T^s along the $T = 200$ and 180 K isotherms, demonstrating that the physics of the Widom line (often approximated by the line of maxima in K_T) can be observed in the properties of water nanodroplets. A maximum in K_T has only recently been observed directly, in experiments on water microdroplets, making our present finding that much more timely.

The referee suggests a plot of K_T versus T at fixed N , and a comparison with bulk values at fixed P . However, such a comparison is complicated by the fact that P is not constant for fixed N (as shown in Fig. 8a). In addition, although our definition of K_T^s is conceptually analogous to K_T , since we calculate K_T^s for a finite subsystem, the values we obtain depend on the size of the subsystem. The plot we have made (Fig. 9) best highlights our key point, that a maximum occurs in K_T^s at low T , similar to that expected near the Widom line in a bulk system.

These additions are implemented on lines 342-359, 595-609, and in Fig. 9.

2. If there is a precursor to a LLCPC under certain conditions would we not expect that fig. 5 shows sharper transition than for bulk water at 1bar. Some discussion of this would be valuable.

This interesting observation is now addressed in the Discussion on lines 385-394. Based on our new analysis of the droplet structure, it is clear that finite-size effects (which round off the discontinuities of a bulk phase transition), as well as strong surface effects (which inhibit the cores of our smaller droplets from reaching the bulk density) both contribute to producing the effect noticed by the referee.

3. Why is there a lower density region inside the droplets under certain condition and not on the surface facing the vacuum. It would be good with some discussion on this point.

To address this important point, we have significantly expanded our analysis and discussion of the radial density profiles of our nanodroplets, and how they change with N and T . As now discussed on lines 208-224, the density maximum of bulk water, the nature of the low-density network that forms in the core at low T , and strong nanoscale surface effects, combine to induce an “inversion” of the density profile as T decreases.

We also quantify the emergence of this unusual density profile using new Figs. 6, 8b, 11 and 14; a new section in Methods (lines 610-620); and we discuss the implications of this change in droplet structure when approaching the LLPT on lines 290-341.

4. In the introduction it is indicated that the freezing temperature decreases for nanodroplets. This could be misleading since what matters is not the freezing temperature but timescales since we are in a metastable region. It is known that nanodroplets have a higher nucleation rate than bulk water and the reason for seeing crystallization at lower temperatures in nanodroplets is related to the much faster cooling rates.

We agree that this point merits clarification, and we have added new text on lines 17-25 of the introduction.

In addition, to check whether the thermodynamic melting temperature is decreased in our model droplets, we have carried out new simulations of small ice nanocrystals that are the same size as some of our liquid droplets. As described in a new subsection (lines 557-573) and figure (Fig. 15) added to Methods, we show that these nanocrystals spontaneously melt well below the melting temperature of the bulk crystal.

Reviewer #2

This paper reports on an extensive simulation study with the TIP4P/2005 model for water of the thermodynamic properties of cold nanodroplets. Cold water nanodroplets occur in Nature and have been studied experimentally. An outstanding question is how their properties are related to that of bulk water. The present study convincingly demonstrates that the bulk equation of state is very well followed even for very small droplets. This result is very significant as it could motivate further experimental studies to access a poorly known region of the phase diagram of water. The main physical effect of the small curvature, the Laplace pressure, is well studied and understood, and this will also be of interest for physicists studying how macroscopic laws transpose to the nanoscale.

The article is very clearly written; all technical details are given and will allow other researchers to reproduce the study. I recommend publication. I have only very few suggestions to slightly improve the context, which the authors may choose to follow or not.

1) In the introduction, the authors give $T_H=232$ K. T_H is an experiment-dependent quantity, see for instance F. Caupin, J. Non-Cryst. Solids 407 (2015) 441448 for a discussion. Indeed, Ref. 13 reports T_H around 227 K for microdroplets, which may be considered bulk. Note however that a recent work on microdroplets evaporating in vacuum, which measured their radii and deduced their temperature [C. Goy et al. to appear in Phys. Rev. Lett. <https://journals.aps.org/prl/accepted/d5076Y88> R5415d6e 36c88093 531548bd9 54946eb2 available on arxiv <https://arxiv.org/abs/1711.02412>] shows that the temperature reported in Ref. 13 might be underestimated, and that the actual temperature might rather be around 230 K.

We agree that the meaning and value of T_H should be clarified in the text, and we thank the referee for suggesting these references. We have adjusted the text on lines 36-41 accordingly, and we have added new references 10 and 18.

2) The authors write experimental efforts increasingly exploit water nanodroplets to probe deeply supercooled conditions for bulk water and cite Refs. 7 and 13. Ref. 7 does study nanodroplets with radii between 3.2 and 5.8 nm, but Ref. 13 rather studies microdroplets, with radii ranging from 4.5 to 18.5 μ m. Other references from the reference list would be more appropriate. A pioneering work on water nanodroplets could also be added: J. Huang, L.S. Bartell, Kinetics of homogeneous nucleation in the freezing of large water clusters, J. Phys. Chem. 99 (1995) 39243931.

These are also useful corrections and clarifications, and we have altered the text on lines 413-416 to acknowledge these points, as well as adding the suggested reference as Ref. 11.

3) Although I understand this is not the focus of the present work, it would be interesting if the authors could compare the values they find for the surface tension and the thickness of the liquid-vapor interface with literature values. They just write γ is increasing with decreasing T , consistent with previous work [25]. Are the values consistent? Is there any effect of curvature (Tolman correction)? See F. Caupin Phys. Rev. E 71, 051605 (2005) and references therein about the thickness of the liquid-vapor interface, and N. Bruot and F. Caupin, Phys. Rev. Lett. 116, 056102 (2016) and references therein about the effect of curvature.

We agree that these are valuable questions to expand on. We have subsequently analyzed our surface tension results in more detail, and find a number of interesting results:

(1) Although the T range in which we evaluate γ lies below that in which γ is estimated in

Ref. 31, our results match well, as shown in the new Fig. 13. As such, our work has extended the estimates for γ to much lower T than has been achieved previously.

(2) Given that our results are obtained from a highly curved nanodroplet surface, and still seem to agree with results for a flat interface near 300 K, we can conclude that the Tolman length for this water model may be approximately zero under the conditions studied. While further work (which we intend to pursue) is required to address the many subtleties involved in estimating the Tolman length, the present results provide an interesting initial estimate.

(3) As shown in Fig. 13, our results for γ increase significantly faster with decreasing T than the extrapolation predicted in Ref. 31. This intriguing behaviour coincides with the emergence of the low-density RTN structure inside our low- T droplets, and suggests that the surface tension is sensitive to this change in liquid structure. There may also be larger curvature effects at lower T .

We intend to pursue a better understanding of the surface tension, including an analysis of the interfacial thickness, in a future work. To address these issues in the current manuscript, we have added a new paragraph on lines 241-257, a new figure (Fig. 13), and Ref. 32.

Reviewer #3

In this work, the authors calculated the density and internal pressure of water nanodroplets with different sizes and temperatures by using TIP4P/2005 water model. The results show a density maximum, which is one of the anomalous behaviors of bulk water, also exists in the central region of water nanodroplets as small as containing a few hundred water molecules. The internal pressure of these droplets, resulted from Laplace pressure, could also be very high for very small droplets. The authors thus argue that the signature of thermodynamic anomalies of bulk water including the precursor of debated liquid-liquid phase transition may also occur in water nanodroplets.

Although this is no doubt an interesting work carried out with great care, I feel the main results/conclusions are quite not unexpected. It is true that water nanodroplets are considered different from bulk water due to both significant surface/volume ratio and large internal pressure, but beneath the molecular layer of those droplets, there exists a bulk-like region in these droplets which behaves like bulk water with a pressure comparable to the corresponding Laplace pressure of droplet. This has been indicated by some previous studies. This work certainly provides a more detailed, rigorous analysis to prove this point, but the main conclusion does not appear as a surprise, but rather reinforcement of what has been speculated.

In the above comment, the referee expresses the concern that in light of previous work, our results are to be expected. After reflecting on this point, we reconsidered and re-analyzed the behaviour observed in our nanodroplets, especially at low T .

Specifically, we have better quantified the “inversion” of the density profile of our droplets as T decreases. While a trend in this direction has been noted in other works [e.g. in Ref. 28], our introduction of Voronoi volumes to evaluate the density profile shows this effect unambiguously for the first time, and shows how dramatic the effect becomes at low T . The existence of a density difference of up to 6% over a distance as small as 1 nm (see the new Fig. 11) is also a remarkable variation for the interior of a nanoscale liquid drop, with potential implications for crystal nucleation and solute behaviour in these nanodroplets.

Furthermore, we re-examined the implications of the deviations we observe between the nanodroplet and the bulk EOS at $T = 200$ and 180 K. If the region inside a droplet beneath the surface layer is bulk-like, then it should attain the density corresponding to that of the bulk at the pressure given by the Laplace pressure P_L . This is exactly what we observe at high T . However, for $T \leq 200$ K and for $200 < N < 1100$, we observe the emergence of a strongly varying density in the nanodroplet interior that never attains the density of the bulk liquid at the Laplace pressure occurring in the droplet (see the new Figs. 6 and 14). That is, as T decreases, our smaller droplets *lose* their bulk-like characteristics. This effect has not been anticipated by, or reported on, in previous work.

We show that these departures from bulk-like behaviour are explained by the interplay between two strong effects at low T and when $P_L < P_c$: (i) the large density gradients in our droplet density profiles, and (ii) the rapid variation in the shape of the bulk EOS near the LLPT. The coupling of these two effects plays a central role in determining the properties of cold water nanodroplets, and has not been identified or discussed in previous work.

The exceptional case of the $N = 100$ droplet is also unexpected, and revealing. As shown

in Fig. 8, a droplet of this size develops sufficient Laplace pressure to enter the regime of the bulk HDL phase for the TIP4P/2005 model. Since the droplet surface and core need not be very different in this regime, bulk behaviour is *not* lost when cooling a droplet of this size, and a strong inverted gradient in the density profile does not emerge. This behaviour confirms that even very small droplets respond to the influence of the EOS for the bulk liquid, including the LLPT.

Consistent with the picture described above, we have directly measured a quantity (K_T^s) analogous to the isothermal compressibility within our droplets, the behaviour of which confirms that our droplets are subject to density fluctuations associated with the approaching critical point of the LLPT (Fig. 9).

In sum, our work provides a detailed description of the pattern of behaviour that occurs for a water-like nanodroplet when the bulk liquid of the same substance has a LLPT. As we discuss in the manuscript, the unanticipated and non-bulk-like behaviours that we observe on cooling provide specific guidance on what to look for in the properties of supercooled water nanodroplets in order to identify the critical region of the LLPT.

Our new emphasis on the non-bulk-like behaviour observed at low T , and its connection to the LLPT, is now discussed at length in a new results section (lines 274-359), and the revised Discussion section (lines 361-425). The new Fig. 6 quantifies the departures from simple droplet behaviour at low T , supplemented by new Figs. 8, 11 and 14, and the new Methods subsection on lines 610-620.

On the other hand, the significance of very small nanodroplets, particularly in hope of resolving the much-debated LLPT, is perhaps also limited. This can be seen by the divergence of EOS between droplets and bulk water at low temperature (Fig. 3). As noted by authors, finite size effect can significantly suppress the density discontinuity near critical point. Overall I think the scope of the paper may be more suitable for a more specialized journal.

Based on the new analysis described above, we feel we have also addressed this concern. As discussed in the Discussion on lines 375-399, there is no doubt that finite-size effects contribute strongly to the EOS deviations observed at low T . However, as emphasized in the revised manuscript, even though the discontinuity near the bulk critical point is absent in the behaviour of nanoscale droplets, several observable phenomena of our droplets can be tied directly to the bulk LLPT, such as: (i) the maximum observed in K_T^s (Fig. 9); (ii) the phase-separation-like density difference observed between the droplet core and surface at low T and for $P_L < P_c$ (Fig. 8b); and (iii) the crossover observed in the density profiles at $P_L = P_c$, where the density profiles are inverted for $P_L < P_c$, and are not inverted for $P_L > P_c$ (Figs. 6, 8b, and 11).

A few minor comments:

1. In both abstract and main text, the authors claim a density maximum occurs in a droplet as small as $N = 300$. This is not totally clear on Fig. 5, as there are only three data points and a maximum is not evident yet. Instead this is clear for $N = 360$.

Our use of $N = 300$ for the onset size is based on an interpolation of our data set to estimate the smallest N at which a density maximum would be observed. However, we agree with

the referee that stating $N = 360$ is probably more clear for the reader, and we have changed the text in the abstract and in the text on line 192 accordingly.

2. What is the error bar for local density in Fig. 4b?

To illustrate the magnitude of the uncertainty in $\rho_v(r)$, we show the errors for several representative cases in the new Fig. 11.

Additional changes, unrelated to Reviewer comments

In our original manuscript, the definition of $\rho_v(r)$ given in the text was not consistent with the method used to calculate the corresponding data. (Both definitions are physically sound, but involve a different order of averaging of the quantities contributing to the ratio of mass and volume.) This has been corrected, resulting in slightly different curves for our $\rho_v(r)$ data. None of our qualitative results are affected by this change.

We have also changed the definition of ρ_c , to be mathematically consistent with the definition of $\rho_v(r)$. Again, the data values have changed only slightly.

Both of these changes also make evaluation of our statistical errors in $\rho_v(r)$ and ρ_c simpler, and these errors are now presented in Figs. 3, 5, 10 and 11.

We also corrected a minor technical error on line 187. “ $N < 360$ ” should have been “ $N \leq 205$ ”.

Reviewer #1 (Remarks to the Author):

The authors have addressed my points in a satisfactory fashion. I recommend publication of the manuscript as it now stands.

Reviewer #2 (Remarks to the Author):

Report on « Thermodynamic anomalies of water nanodroplets »

I have read the revised manuscript and the response to the three reports. The Authors have addressed all issues raised, and have even performed additional simulations. This results in an improved manuscript. In particular, the expanded discussion of the radial density profile, the estimate of the surface tension and of a proxy for the isothermal compressibility, are highly valuable additions. I am convinced about the significance of this work for our understanding of supercooled water and nanoscale phenomena.

I would just like to mention minor details that could be improved before publication.

1) The second paragraph of the main text discusses the homogeneous nucleation temperature T_H (lines 36-41). In response to item 1 of my first report, the Authors have added that its “precise value depends on the experimental protocol [10, 15–18]”. This addresses my comment appropriately. However, I have now noticed another issue with the end of the sentence: “frustrating attempts to directly observe the LLPT predicted to occur at lower T ”. Depending on the prediction, the LLPT might exist only at sufficiently high pressure (e.g. above 170 MPa for TIP4P/2005 water), and not at ambient pressure. Here, the range of T_H values given in the sentence corresponds to experiments at ambient pressure. At higher pressure, T_H is lower (see e.g. Kanno, Speedy and Angell, Science, 1975, 189, 880-881). Still, the argument about crystallization preventing the observation of a LLPT holds. Therefore, to avoid any misunderstanding, I suggest a very slight modification of the sentence as follows:

“However, bulk samples of liquid water crystallize at a homogeneous nucleation temperature T_H (encountered at ambient pressure in the range 227-232 K, where the precise value depends on the experimental protocol [10, 15–18]), frustrating attempts to directly observe the LLPT predicted to occur at lower T .”

2) As the simulations are of course performed for a discrete set of values of N and T , when a change of behavior is observed at a threshold value, it is better to include it in the description, using the sign “greater than or equal to” rather than “strictly greater than” (or equivalently “less than or equal to” rather than “strictly less than”). For instance, on line 219, “ $N \geq 200$ ” rather than “ $N > 100$ ”, and on line 286, “ $N \leq 776$ ” rather than “ $N < 1100$ ”.

3) In the new discussion of isothermal compressibility, the Authors write (lines 353-355): “Experiments on water microdroplets have recently produced the first direct observations of a KT maximum in supercooled water [17].” I would like to stress that, in the cited reference, the observation is not direct, as it involves analysis of low angle x-ray scattering data to extract the $q=0$ limit of $S(q)$, and then KT , which requires several assumptions, as for instance the extrapolation of density data to temperatures below which they were measured. In addition, it is not the first report, as an earlier report of a KT maximum in doubly metastable water (simultaneously supercooled and stretched) is available (Holten et al. J. Phys. Chem. Lett. 2017, 8, 5519-5522), which used microdroplets of water trapped in a quartz matrix.

4) In response to item 2 of my first report, the Authors have modified lines 413-416 to cite appropriate references. Still, on line 416, Ref. 18 should be added to Refs. 16 and 17 about microdroplets. For nanodroplets, a new work (Amaya and Wyslouzil, J. Chem. Phys. 2018, 148, 084501) has appeared since my first report, and could also be cited.

Reviewer #3 (Remarks to the Author):

In revision, the authors made efforts to address all the raised questions/concerns. I find most of the new results and analyses very useful, the paper significantly improved overall. Although surface effect and finite size effect cannot be explicitly distinguished for very small water clusters at low T , where this reviewer still has some concerns about their relevance to LLPT, I think the revised paper may be publishable overall.

Reviewer #1

The authors have addressed my points in a satisfactory fashion. I recommend publication of the manuscript as it now stands.

No changes requested.

Reviewer #2

I have read the revised manuscript and the response to the three reports. The Authors have addressed all issues raised, and have even performed additional simulations. This results in an improved manuscript. In particular, the expanded discussion of the radial density profile, the estimate of the surface tension and of a proxy for the isothermal compressibility, are highly valuable additions. I am convinced about the significance of this work for our understanding of supercooled water and nanoscale phenomena.

I would just like to mention minor details that could be improved before publication.

1) The second paragraph of the main text discusses the homogeneous nucleation temperature TH (lines 36-41). In response to item 1 of my first report, the Authors have added that its precise value depends on the experimental protocol [10, 1518]). This addresses my comment appropriately. However, I have now noticed another issue with the end of the sentence: frustrating attempts to directly observe the LLPT predicted to occur at lower T. Depending on the prediction, the LLPT might exist only at sufficiently high pressure (e.g. above 170 MPa for TIP4P/2005 water), and not at ambient pressure. Here, the range of TH values given in the sentence corresponds to experiments at ambient pressure. At higher pressure, TH is lower (see e.g. Kanno, Speedy and Angell, Science, 1975, 189, 880-881). Still, the argument about crystallization preventing the observation of a LLPT holds. Therefore, to avoid any misunderstanding, I suggest a very slight modification of the sentence as follows:

“However, bulk samples of liquid water crystallize at a homogeneous nucleation temperature TH (encountered at ambient pressure in the range 227-232 K, where the precise value depends on the experimental protocol [10, 1518]), frustrating attempts to directly observe the LLPT predicted to occur at lower T.”

We agree with the Reviewer’s comment, and have changed the text on lines 36-41 according to the suggested wording given above.

2) As the simulations are of course performed for a discrete set of values of N and T, when a change of behavior is observed at a threshold value, it is better to include it in the description, using the sign “greater than or equal to” rather than “strictly greater than” (or equivalently “less than or equal to” rather than “strictly less than”). For instance, on line 219, “ $N \geq 200$ ” rather than “ $N > 100$ ”, and on line 286, “ $N \leq 776$ ” rather than “ $N < 1100$ ”.

This is also a useful clarification, and we have implemented it on lines 220, 286, 297, 307 and 310.

3) In the new discussion of isothermal compressibility, the Authors write (lines 353-355): Experiments on water microdroplets have recently produced the first direct observations of a KT maximum in supercooled water [17]. I would like to stress that, in the cited reference, the observation is not direct, as it involves analysis of low angle x-ray scattering data to extract the $q=0$ limit of $S(q)$, and then KT , which requires several assumptions, as for instance the extrapolation of density data to temperatures below which they were measured. In addition, it is not the first report, as an earlier report of a KT maximum in doubly metastable water (simultaneously supercooled and stretched) is available (Holten et al. J. Phys. Chem. Lett. 2017, 8, 5519-5522), which used microdroplets of water trapped in a quartz matrix.

We agree with this comment, and we have modified the text on lines 353-357 referring to Ref. 17 accordingly. We have also added the suggested citation as Ref. 33.

4) In response to item 2 of my first report, the Authors have modified lines 413-416 to cite appropriate references. Still, on line 416, Ref. 18 should be added to Refs. 16 and 17 about microdroplets. For nanodroplets, a new work (Amaya and Wyslouzil, J. Chem. Phys. 2018, 148, 084501) has appeared since my first report, and could also be cited.

On line 418, we have added Ref. 18, as well as the suggested new citation as Ref. 28.

Reviewer #3

In revision, the authors made efforts to address all the raised questions/concerns. I find most of the new results and analyses very useful, the paper significantly improved overall. Although surface effect and finite size effect cannot be explicitly distinguished for very small water clusters at low T , where this reviewer still has some concerns about their relevance to LLPT, I think the revised paper may be publishable overall.

No changes requested.